# Algorithmic Design of an FPGA-Based Calculator for Fast Evaluation of Tsunami Wave Danger

**Mikhail Lavrentiev [1,\*], Konstantin Lysakov [1], Andrey Marchuk [1,2], Konstantin Oblaukhov [1] and Mikhail Shadrin [1]**

[1] Institute of Automation and Electrometry SB RAS, 630090 Novosibirsk, Russia; lysakov@sl.iae.nsk.su (K.L.); mag@omzg.sscc.ru (A.M.); oblaukhov.konstantin@gmail.com (K.O.); mikesha@sl.iae.nsk.su (M.S.)

[2] Institute of Computational Mathematics and Mathematical Geophysics SB RAS, 630090 Novosibirsk, Russia

\* Correspondence: mmlavrentiev@gmail.com; Tel.: +7-913-925-5469

**Abstract:** Events of a seismic nature followed by catastrophic floods caused by tsunami waves (the incidence of which has increased in recent decades) have an important impact on the populations of littoral regions. On the coast of Japan and Kamchatka, it takes nearly 20 min for tsunami waves to approach the nearest dry land after an offshore seismic event. This paper addresses an important question of fast simulation of tsunami wave propagation by mapping the algorithms in use in field-programmable gate arrays (FPGAs) with the help of high-level synthesis (HLS). Wave propagation is described by the shallow water system, and for numerical treatment the MacCormack scheme is used. The MacCormack algorithm is a direct difference scheme at a three-point stencil of a "cross" type; it happens to be appropriate for FPGA-based parallel implementation. A specialized calculator was designed. The developed software was tested for precision and performance. Numerical tests computing wave fronts show very good agreement with the available exact solutions (for two particular cases of the sea bed topography) and with the reference code. As the result, it takes just 17.06 s to simulate 1600 s (3200 time steps) of the wave propagation using a $3000 \times 3200$ computation grid with a VC709 board. The step length of the computational grid was chosen to display the simulation results in sufficient detail along the coastline. At the same time, the size of data arrays should provide their free placement in the memory of FPGA chips. The rather high performance achieved shows that tsunami danger could be correctly evaluated in a few minutes after seismic events.

**Keywords:** algorithm implementation; high-level synthesis; tsunami wave; numerical modeling



## 1. Introduction

In such matters as timely notification of the populations of coastal territories about the threat of a tsunami (the predicted wave height on a given section of coastline), it is very important to obtain the results of the analysis as soon as possible. After the Great Tohoku Earthquake of 11 March 2011, a destructive tsunami wave reached the nearest coastline ~20 min after the seismic event. This imposes particularly stringent requirements on data processing performance for tsunami danger prediction. Currently, several software packages are used for numerical modeling in different countries, the best known of which are MOST (Method of Splitting Tsunamis, NOAA Pacific Marine Environmental Laboratory, Seattle, USA), COMCOT (Cornell University, USA; NS Science, New Zeland), TUNAMI-N1/TUNAMI-N2 (Tohoku University, Japan), VOLNA, and NAMI-DANCE (TSUNAMI Modeling Software by Special Bureau of Sakhalin, Russia and METU Turkey). In order to obtain results in the shortest time, some algorithms are implemented on GPUs and high-performance clusters. However, none of these packages gives a result in the required timeframe of 1–2 min.

Since power supply outages are possible in the event of catastrophic earthquakes, it is important to be able to quickly receive and analyze the measured data received, regardless of the centralized power supply. The FPGA-based calculator previously proposed by the authors [1] makes it possible to fulfill both of these requirements, which usually contradict

one another. The use of HLS technology can significantly reduce the porting time of the algorithms used to the proposed hardware platform. In some cases, HLS makes possible the easily adaptation of the existing implementations of some algorithms to solve data processing problems related to tsunami danger prediction.

In this paper, the aforementioned advantages of HLS technology are demonstrated by the example of a convolution algorithm widely used in image processing. As it happens, the MacCormack finite-difference scheme for the numerical solution of the shallow water equation system (a well-established, commonly accepted model for calculating the propagation of a tsunami wave off the coastline) is quite similar to the aforementioned convolution algorithm. This makes it possible in a short time to adapt an existing FPGA-based software/hardware complex for the numerical solution of a system of shallow water equations. An acceleration of ~300 times (for one time step) was achieved compared to single-threaded computing. This comparison was made against a non-optimized, single-threaded reference MOST algorithm running on a 3 GHz CPU. We also compared the FPGA-based solution against a reference MacCormack implementation, parallelized using OpenMP and SIMD instructions on a modern CPU (Intel Core i9-9900K, 8 cores running 4 GHz); the FPGA-based solution was 11.8 times faster. Essentially, this allows us to solve the problem of the timeliness of calculating the wave parameters along the coastline, since the counting time with a grid of 3000 × 3200 nodes takes only 38.4 s to simulate 1 h (7200 time steps) of the wave propagation.

To clarify the use of hardware/software terminology, we note that all computations were mapped to hardware, while software was used to organize access to data and save results. Data were stored on dynamic RAM (DDR3/4) attached to the FPGA. Communication was via internal interconnect in the case of Zynq SoC, or PCIe in the case of VC709, using a DMA engine on the FPGA and driver software on the host.

This paper is presented in the following sequence: Section 2.1 briefly describes the standard implementation of the convolutional algorithm. Then, in Section 2.2, the sequence of steps required to efficiently transfer the algorithm to an FPGA is described, and in Section 2.3, the final implementation using Xilinx's Vitis HLS technology is given. Section 2.4 compares the results with the low-level implementation of the algorithm in Verilog language [2]. Finally, in Section 2.5, the MacCormack finite-difference scheme for the numerical solution of the system of shallow water equations [3,4] is given. We also indicate features of the MacCormack scheme that enable the application of HLS technology [5] for the construction of computing pipelines to adapt the mentioned implementation of the convolutional algorithm. Section 3.1 is devoted to the description of testing the proposed hardware–software solution (the FPGA-based calculator) by comparing the numerical solutions obtained for the model cases of bathymetry with the exact solutions of the shallow water equation system in the ray approximation [6,7] known in the literature, as well as with the results obtained using the MOST software package [8]. Section 3.2 shows the results of computational experiments conducted with the developed calculator to calculate the distribution of maximum heights of tsunami waves from the reconstructed source of the catastrophic tsunami on 11 March 2011, caused by the Great Tohoku Earthquake off the northeast coast of Japan. In the final section, we discuss the advantages of HLS, using simulation of tsunami wave propagation as a practical example.

## 2. Materials and Methods

### 2.1. Problem Statement

One of the illustrative examples of using HLS to simplify the development process is the implementation of convolutional (matrix) filters for image processing. Similar operations are also used in convolutional neural networks.

A two-dimensional array with dimensions (*nx, ny*) representing the luminance (*Y*) values as single precision floating point numbers is given. For each pixel of the image, it is necessary to arrange its convolution with neighboring pixels using the given weight matrix (convolution kernel):

$$g(x,y) = w * f(x,y) = \sum_{dx=-a}^{a} \sum_{dy=-a}^{a} w(dx,dy) f(x+dx, y+dy)$$

where $g(x, y)$ is the resulting pixel with coordinates $(x, y)$, $f(x,y)$ is the original image, $w(dx, dy)$ is the convolution kernel (size $NxN$), and $N = a*2 + 1$ is the size of the convolution kernels. There are no data for convolution at the border of the image, so there are several options for padding the border with pixels. For this article, we implemented convolution with a $3 \times 3$ kernel and padded the border with the edge pixels.

Listing 1 of the algorithm's realization in C/C++ language is given below.

**Listing 1.** Reference implementation of the convolutional filter.

```
void filter (const Pixel * input, int width, int height,
          Coeff kernel[KERNELSIZE][KERNELSIZE], Pixel * output) {
     for (int y = 0; y < height; y++) {
          for (int x = 0; x < width; x++) {
               Pixel result = 0.0;
               for (int ky = 0; ky < KERNELSIZE; ky++) {
                    for (int kx = 0; kx < KERNELSIZE; kx++) {
                         result += getPixel(input, width, height,
                                             x, y, kx, ky)
                                   * kernel[ky * KERNELSIZE + kx];
                    }
               }
               output[y * width + x] = result;
          }
     }
}
```

Here, Pixel and Coeff are synonyms for the type of float, the kernel size is set by the constant KERNELSIZE, and getPixel means to obtain the pixel of the original image taking into account the padding of borders. The results are given in Listing 2 below.

**Listing 2.** Pixel sampling.

```
inline Pixel getPixel (const Pixel * input, int width, int height,
                          int x, int y, int kx, int ky) {
     int ix = x + kx − KERNELSIZE/2;
     int iy = y + ky − KERNELSIZE/2;
     ix = (ix < 0) ? 0: ((ix ≥ width) ? (width-1): ix);
     iy = (iy < 0) ? 0: ((iy ≥ height) ? (height-1): iy);
     return input[iy * width + ix];
}
```

The described algorithm was treated as the reference algorithm, on the basis of which we developed an implementation on an FPGA.

### 2.2. Pipeline (Stream) Algorithm

The reference algorithm assumes that the original image is stored in random-access memory. However, in the case of hardware implementation, this is not always the case; more often the image is a stream, the pixels come in turns, the image is not stored anywhere in its entirety, and the processor has no way to obtain an arbitrary pixel. In this case, it is necessary to transform the algorithm by making it stream-oriented; this is the first step to implementation on an FPGA, and can be achieved as shown in Listing 3:

**Listing 3.** Stream-based convolutional filter.

```
void filter(std::istream & input, int width, int height,
  Coeff kernel[KERNELSIZE][KERNELSIZE], std::ostream & output){
  int ox = -KERNELSIZE/2, oy = -KERNELSIZE/2;
  int ix = 0, iy = 0;
  Pixel stencil[KERNELSIZE][MAXWIDTH];
  for(int i = 0; i < width*(height + KERNELSIZE/2) + KERNELSIZE/2;
                                                              i++) {

    if (iy < height)
      stencil[iy % KERNELSIZE][ix] = readPixel (input);
    if ((ox ≥ 0) && (oy ≥ 0)) {
      Pixel result = 0.0;
      for (int ky = 0; ky < KERNELSIZE; ky++) {
        for (int kx = 0; kx < KERNELSIZE; kx++) {
          int sx = ox + kx − KERNELSIZE/2;
          int sy = oy + ky − KERNELSIZE/2;
          sx = (sx < 0) ? 0: ((sx ≥ width) ? (width-1): sx);
          sy = ((sy < 0) ? 0: ((sy ≥ height) ? (height-1): sy))
                                                              %KERNELSIZE;
          result += stencil[sy][sx] * kernel[ky][kx];
        }
      }
      WritePixel (output, result);
    }
    bool ixend = (ix == width − 1);
    ix = ixend ? 0: ix + 1;
    iy = ixend ? (iy + 1): iy;
    bool oxend = (ox == width − 1);
    ox = oxend ? 0: ox + 1;
    oy = oxend ? (oy + 1): oy;
  }
}
```

Here, the pixels come sequentially through the streams of the C++ language. The algorithm remembers the previous lines of the image in the *stencil* array. As the pixels arrive, they are convolved with the pixels stored in the *stencil* array. In total, the KERNELSIZE number of lines must be stored. Since the data required for the convolution of the first pixel will appear only when the KERNELSIZE line arrives, such a streaming algorithm has a delay.

## 2.3. HLS-Based Algorithm Porting

Porting an existing algorithm from C/C++ in the Vivado/Vitis HLS environment consists of creating directives to the synthesizer indicating how one or another language construct should be implemented. If one starts the synthesis of the original streaming algorithm, the synthesizer will generate an ALU with multiplication and addition operations, as well as a state machine that will read data from the input interfaces and load them into the ALU in the desired order. This approach uses only two operational blocks, but requires several clock cycles to compute one pixel.

In real tasks, it is necessary to use pipelining in order to be able to process one pixel per cycle; for this, the following PIPELINE directive given in Listing 4 is used.

**Listing 4.** Vitis HLS pipelining.

```
for (int i = 0; i < width*(height + KERNELSIZE/2) + KERNELSIZE/2; i++) {
    #pragma HLS PIPELINE
    . . .
}
```

This tells the synthesizer that this loop should be implemented as a pipeline; each iteration should start as soon as possible (interval, II)—ideally, every clock cycle. Not every loop can be perfectly pipelined; however, in our case, the synthesizer successfully pipelined the loop at intervals of 1 (every clock cycle).

However, the resulting processor uses 9 times more memory for *stencil* than needed. The source of this problem is suboptimal access to the *stencil* variable. The fact is that the memory inside the FPGA has only two access ports; by default, HLS generates only one and, accordingly, the processor can access only one value from the *stencil* at once, and convolution of each pixel requires access to 9 (KERNELSIZE * KERNELSIZE) values at the same time. The synthesizer bypasses this limitation by creating 9 copies of the *stencil* buffer.

However, if one looks closer at the algorithm, it become apparent that, for each iteration, one needs only to access one value from the previous lines, and the values of the previous pixels can be saved between iterations. Moreover, the very first line in the buffer can be immediately replaced with the current one. Then, only *KERNELSIZE-1* samples from the stencil are required. Moreover, it should be noted that for each line of the buffer, only one pixel is selected at a time, so the buffer can be split line by line between memory blocks, providing the required number of ports. Thus, Listing 5 of the algorithm is as follows:

**Listing 5.** Convolutional filter with array partitioning.

```
...
Pixel stencil[KERNELSIZE-1][MAXWIDTH];
#pragma HLS ARRAYPARTITION dim = 1 type = complete variable = stencil
Pixel pixels[KERNELSIZE][KERNELSIZE];
#pragma HLS ARRAYPARTITION dim = 1 type = complete variable = pixels
#pragma HLS ARRAYPARTITION dim = 2 type = complete variable = pixels
for (int i = 0; i < width*(height + KERNELSIZE/2) + KERNELSIZE/2; i++) {
    #pragma HLS PIPELINE
    Pixel cur = 0.0f;
    if (iy < height)
        cur = input.read();
    for (int ky = 0; ky < KERNELSIZE; ky++) {
        #pragma HLS UNROLL
        for (int kx = 0; kx < KERNELSIZE-1; kx++) {
            #pragma HLS UNROLL
            pixels[ky][kx] = pixels[ky][kx + 1];
        }
        if (ky != KERNELSIZE-1)
            pixels[ky][KERNELSIZE-1]=
                                stencil[(iy + ky)%(KERNELSIZE-1)][ix];
        else
            pixels[ky][KERNELSIZE-1] = cur;
    }
    stencil[iy % (KERNELSIZE-1)][ix] = cur;
    if ((ox ≥ 0) && (oy ≥ 0)) {
        Pixel result = 0.0;
        for (int ky = 0; ky < KERNELSIZE; ky++) {
            #pragma HLS UNROLL
            for (int kx = 0; kx < KERNELSIZE; kx++) {
                #pragma HLS UNROLL
                int sx = ox + kx − KERNELSIZE/2;
                int sy = oy + ky − KERNELSIZE/2;
                sx = (sx < 0) ? (KERNELSIZE/2-ox): ((sx ≥ width) ?
                                        (width-1 + KERNELSIZE/2-ox): kx);
                sy = (sy < 0) ? (KERNELSIZE/2-oy): ((sy ≥ height) ?
                                        (height-1 + KERNELSIZE/2-oy): ky);
                result += pixels[sy][sx] * kernel[ky][kx];
            }
        }
        output. write (result);
    }
...
```

Here, the *pixels* array is used to store the current pixels participating in the convolution. *HLS ARRAYPARTITION* specifies how the array is decomposed across FPGA memory blocks. *HLS UNROLL* unrolls a loop into linear code at the compilation stage.

*2.4. Comparison to Verolog Realization*

Using a similar approach, the filtering algorithm was implemented in the Verilog language. Since the Verilog synthesizer does not have standard operations on floating point numbers, ready-made libraries (IP cores) were used as part of Xilinx *Vivado*. Unfortunately, the Verilog language is not sufficient to write universal parameterizable code, so part of the module had to be implemented in a fixed manner, for a kernel size equal to 3.

The HLS system allows synthesis with the setting of the required maximum frequency, which allows one to adjust the maximum frequency/size ratio of the final circuit. For Verilog code, one must modify the code to increase the frequency. In fact, there are

alternative options to increase the frequency. However, RTL synthesis and implementation are very limited in what they can improve within design, because structure/behavior is strictly specified. For example, they cannot increase pipeline depth automatically to achieve better timing. HLS tools, on the other hand, are free to choose how to implement the desired algorithm.

In addition, using the C++ language and HLS, it is possible to easily replace the float type with half-precision or fixed-point numbers by changing exactly one line. However, we did not use this option for tsunami-related applications.

The HLS code was synthesized for 148.5 MHz (1080p50 video frequency) and 400 MHz (close to the theoretical FPGA limit). The final synthesis took place for the Zynq UltraScale + (xczu7ev) platform. All solutions were analyzed for maximum frequency using Vivado tools. Strictly speaking, the maximum frequency is not explicitly given in Vitis/Vivado timing reports. Vivado HLS can report Fmax, calculated from post-synthesis timing reports (using the "target period-worst slack" formula); the same was done for the Verilog design. Target frequency (specified in constraints) was the same as the target frequency for the HLS design (400 MHz and 148.5 MHz), and 148.5 MHz for Verilog, which is not a real frequency on which the implemented design can run.

All of the algorithms on any platform operate on data in single-precision floating point (IEEE 754) format; no conversions to fixed points or half-precision were made. Such conversion may result in better FPGA resource utilization, but requires deep data analysis for every case, and is beyond our consideration here.

The final results are presented in the table below.

As can be seen from Table 1, in the example of real tasks, the amount of program code was reduced by 5–8 times, while the implementation performance (clock frequency of operation) was maintained. However, the side-effect was an increase in the number of logical blocks (LUT) and arithmetic processing blocks (DSP). The increased memory block consumption in the naïve HLS example demonstrates that in order to use HLS effectively, one must know the FPGA's structure and features well.

**Table 1.** Comparison of FPGA-based implementations by resource utilization.

| Implementation | Lines of Code | Maximum Frequency | Number of LUTs | Number of Flip-Flops | Number of BRAMs (36 K) | Number of DSP Blocks |
|---|---|---|---|---|---|---|
| Naïve HLS | ~40 | 308 MHz | 7364 | 7467 | 198 | 48 |
| HLS (148.5 MHz) | ~60 | 308 MHz | 4575 | 5100 | 15 | 39 |
| HLS (400 MHz) | ~60 | 637 MHz | 6145 | 9717 | 16 | 30 |
| HLS half (400 MHz) | ~60 | 514 MHz | 2875 | 4261 | 8 | 30 |
| Verilog | ~280 | 374 MHz | 4302 | 6709 | 15 | 34 |

### 2.5. Tsunami Math Model and Numerical Scheme

To simulate the tsunami wave propagation, the nonlinear shallow water PDE system [9] was used.

$$
\begin{aligned}
H_t + (uH)_x + (vH)_y &= 0, \\
u_t + uu_x + vu_y + gH_x &= gD_x \\
v_t + uv_x + vv_y + gH_y &= gD_y,
\end{aligned} \tag{1}
$$

where $H(x,y,t) = \eta(x,y,t) + D(x,y)$ is the distance from the sea's surface to the bottom, $\eta$ is the sea surface displacement relative to the mean sea level (wave height), $D(x,y)$ is the water depth (which is supposed to be known at all grid points), $u$ and $v$ are the water flow velocity components in the $x$ and $y$ directions, respectively, and $g$ is the gravity force acceleration. Note that the lower indices $t$, $x$, $y$ in (1) indicate partial derivatives with respect to time and space variables.

For numerical experiments, the following finite-difference MacCormack scheme [3] was implemented:

$$\frac{\widehat{H}_{ij}^{n+1} - H_{ij}^n}{\tau} + \frac{H_{ij}^n u_{ij}^n - H_{i-1j}^n u_{i-1j}^n}{\Delta x} + \frac{H_{ij}^n v_{ij}^n - H_{ij-1}^n v_{ij-1}^n}{\Delta y} = 0$$

$$\frac{\widehat{u}_{ij}^{n+1} - u_{ij}^n}{\tau} + u_{ij}^n \frac{u_{ij}^n - u_{i-1j}^n}{\Delta x} + v_{ij}^n \frac{u_{ij}^n - u_{ij-1}^n}{\Delta y} + g\frac{\eta_{ij}^n - \eta_{i-1j}^n}{\Delta x} = 0$$

$$\frac{\widehat{v}_{ij}^{n+1} - v_{ij}^n}{\tau} + u_{ij}^n \frac{v_{ij}^n - v_{i-1j}^n}{\Delta x} + v_{ij}^n \frac{v_{ij}^n - v_{ij-1}^n}{\Delta y} + g\frac{\eta_{ij}^n - \eta_{ij-1}^n}{\Delta y} = 0$$

$$\frac{H_{ij}^{n+1} - \left(\widehat{H}_{ij}^{n+1} + H_{ij}^n\right)/2}{\tau/2} + \frac{\widehat{H}_{i+1j}^{n+1}\widehat{u}_{i+1j}^{n+1} - \widehat{H}_{ij}^{n+1}\widehat{u}_{ij}^{n+1}}{\Delta x} + \frac{\widehat{H}_{ij+1}^{n+1}\widehat{v}_{ij+1}^{n+1} - \widehat{H}_{ij}^{n+1}\widehat{v}_{ij}^{n+1}}{\Delta y} = 0$$

$$\frac{u_{ij}^{n+1} - \left(\widehat{u}_{ij}^{n+1} + u_{ij}^n\right)/2}{\tau/2} + u_{ij}^n \frac{\widehat{u}_{i+1j}^{n+1} - \widehat{u}_{ij}^{n+1}}{\Delta x} + v_{ij}^n \frac{\widehat{u}_{ij+1}^{n+1} - \widehat{u}_{ij}^{n+1}}{\Delta y} + g\frac{\widehat{\eta}_{i+1j}^{n+1} - \widehat{\eta}_{ij}^{n+1}}{\Delta x} = 0$$

$$\frac{v_{ij}^{n+1} - \left(\widehat{v}_{ij}^{n+1} + v_{ij}^n\right)/2}{\tau/2} + u_{ij}^n \frac{\widehat{v}_{i+1j}^{n+1} - \widehat{v}_{ij}^{n+1}}{\Delta x} + v_{ij}^n \frac{\widehat{v}_{ij+1}^{n+1} - \widehat{v}_{ij}^{n+1}}{\Delta y} + g\frac{\widehat{\eta}_{ij+1}^{n+1} - \widehat{\eta}_{ij}^{n+1}}{\Delta y} = 0$$

where $H_{ij}^n$, $u_{ij}^n$, and $v_{ij}^n$ are the gridded variables that correspond to the $H$, $u$, and $v$ functions, respectively, in Differential System (1). The parameters $\tau$, $\Delta x$, and $\Delta y$ are the time step and spatial steps of the computational grid. In order to account for the spherical shape of the Earth, we used a decreasing grid step with respect to longitude for larger values of latitude. The notation $F_{ij}^n$ represents variables at time layer $n$, $\widehat{F}_{ij}^{n+1}$ represents intermediate values, and $F_{ij}^{n+1}$ corresponds to the variables at time layer $n + 1$.

In order to calculate wave parameters $H_{ij}$, $u_{ij}$ and $v_{ij}$ for the time step $n + 1$ along the line $j = 2$ of the gridded arrays, we need the data from the time step $n$ along the lines $j = 1,2,3$. Wave parameters along the line $j = 1$ are obtained using the boundary conditions. When the wave parameters for the time step $n + 1$ are already calculated for at least 3 neighboring lines $j = 2,3,4$, the calculator #3 can start working to obtain the wave parameters for the time step $n + 2$ along the line $j = 3$, and so on (Figure 1). All 8 calculators work simultaneously on wave parameters for 8 time layers. The inner memory of each calculator keeps all data (wave parameters and bathymetry) for one time layer.

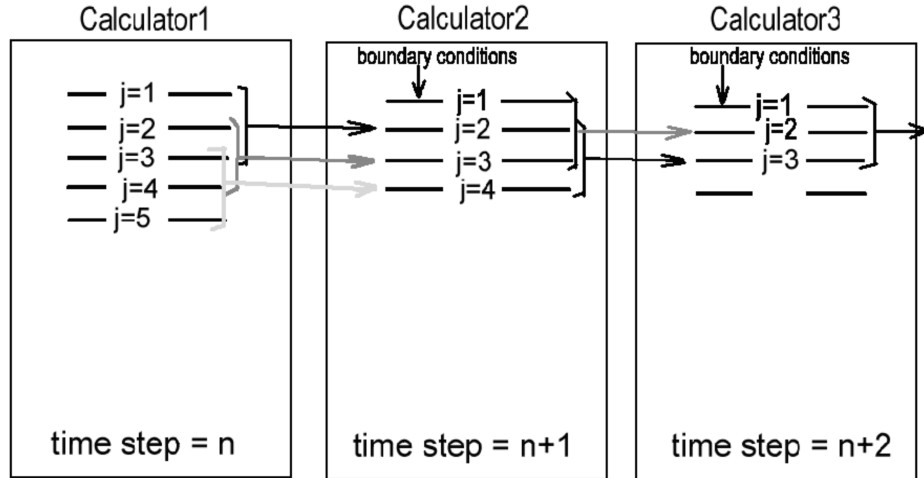

**Figure 1.** The scheme of computing the conveyer (pipeline) using 3+ calculators.

In fact, this scheme is similar in structure to the convolutional algorithm described above. Thus, the proposed approach was applied to the MacCormack scheme. As the result, a special processor element (PE) was designed to implement one step of the algorithm (see [1]). By composing PEs into a line, the calculator can be scaled to different FPGAs. Below we present an implementation of the MacCormack algorithm in C using HLS. The design is intended for a ZCU106 board (FPGA xczu7ev) and 400 MHz. Xczu7ev has 230,400 LUTs, 460,800 flip-flops, 312 + 512 BRAM blocks (36 K), and 1728 DSP blocks.

The final PE has the following parameters, collected in Table 2:

**Table 2.** Resource utilization for MacCormack implementation.

| Implementation | Lines of Code | Maximum Frequency | Number of LUTs | Number of Flip-Flops | Number of BRAMs (36 K) | Number of DSP Blocks |
|---|---|---|---|---|---|---|
| MacCormack | ~280 | 462 MHz | 29,242 | 49,520 | 64 | 189 |

Once again, we would like to stress that all of the algorithms operate on data in single-precision floating points (IEEE 754); no conversions to fixed points or half-precision were made. The entire data array (D, η, u, v) was stored in dynamic RAM (DDR) placed near the FPGA chip on the board. Initial data were transferred to the FPGA from the PC using the PCIe interface, or internal interconnect in the case of an FPGA + CPU solution, such as Xilinx Zynq. The maximum acceptable dataset size depends on the size of the dynamic RAM. For a VC709 accelerator, the maximum dataset is 16,384 × 16,384 points of bathymetry. The bathymetry size for our experiments was 3000 × 3200, for practical reasons (size of resulting data, computation time on FPGA/CPU, etc.).

## 3. Results

### 3.1. Numerical Tests at Model Bathymetry

In order to test the developed numerical algorithm in the area of 1000 × 1000 calculation grid points, a simple example of bottom topography was used, representing a uniform bottom slope ($D(x,y) = ky$) and a parabolic bottom profile ($D(x,y) = ky^2$), where the parameter $y$ represents the distance from the rectilinear coastline. With the spatial steps of the grid in both directions equal to 1000 m, the coefficient $k$ was chosen so that the depth varied from 0 m at the lower boundary (coastline) to 10,000 m at the upper boundary of the computation domain (see Figure 2).

When modeling the tsunami propagation via digital bathymetry measuring 3000 × 3200 calculated nodes, approximation of the real bottom topography of the area was conducted off the northeast coast of the Japanese island of Honshu. This area located between 34 and 42 degrees NL (northern latitude), as well as between 140 and 147.46 degrees EL (eastern longitude). The digital depth array for this area was built on the basis of oceanographic data from the JODC [10]. The step length of the calculated grid, linked to geographical coordinates, was 0.0025 arc degrees, which is equal to 278 m in the north–south direction and 219 m in the west–east direction.

To assess the reliability and accuracy of the proposed method, numerical calculations were carried out on model topographies of the bottom, for which an exact solution was obtained via ray approximation [6,7]. In these test calculations, the bottom relief of which is described in the previous subsection, the tsunami propagation was modeled from a circular initial disturbance area with a radius of $r_0$ = 50 km, with the center located at a distance of 300 km from the coastline, and at the same distance from the lateral boundaries of the region. In Figure 2, the area of the initial displacement of the water surface is filled with yellow. The vertical displacement of the water surface in the source was given by the following formula:

$$\eta = \eta_0 \left( 1 + \cos\left( \frac{\pi \cdot r}{r_0} \right) \right),$$

where $r$ represents the distance from the center of the initial water surface elevation ($r \leq r_0$). In Figure 2a,b, showing the results of testing on model bottom reliefs, the upward

direction coincides with the offshore direction. The propagation of the wave generated by the described round-shaped source was simulated using MOST software, and by the MacCormack difference scheme implemented on an FPGA. According to the results of calculations for each of the schemes, isolines of distributions of wave height's maxima over the sloping bottom (Figure 2a) and at the parabolic bottom topography (Figure 2b) were visualized.

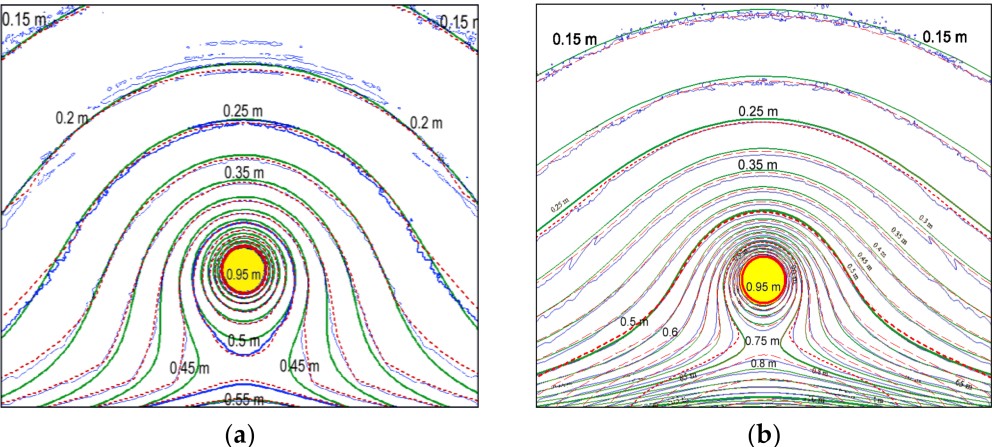

**(a)**      **(b)**

**Figure 2.** Comparison of the wave height maxima distributions for the exact solution (green); results obtained via the proposed FPGA-based method (red dashed lines) and MOST single-precision results (blue). (**a**) The comparison results for sloping bottom topography; (**b**) the comparison results for the parabolic bottom topography.

### 3.2. Numerical Results

To simulate a tsunami on real bathymetry, the tsunami source proposed in [11] was taken as the initial vertical displacement of the water surface; its location in the computational domain is shown in Figure 3 in the form of isolines of the vertical displacement field.

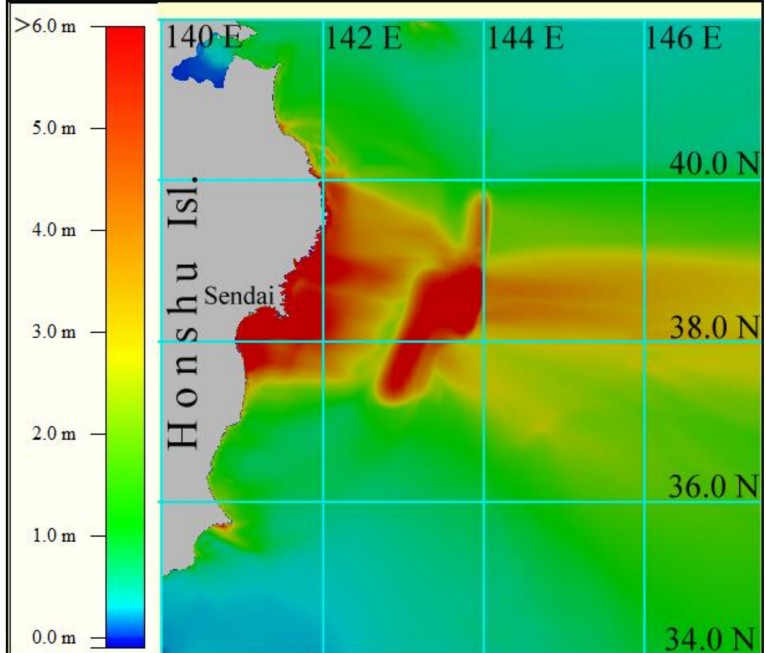

**Figure 3.** Digital bathymetry and tsunami source of the 11 March 2011 event. The isolines of the field of vertical elevation of the water surface at an interval of 1 m are drawn in white, and the isolines of the initial abatement of water level are drawn in black. The source is outlined by ±0.05 m level lines. The maximum elevation in the source is limited to +9 m, and the lowest value is equal to −4 m.

Based on the results of numerical calculation of the propagation of the tsunami wave generated by the source shown in Figure 3, the estimates of maximum tsunami height in the entire computational domain and along the coast were obtained (see Figure 4).

The obtained numerical results show that for the tsunami source considered (see Figure 3), the maximum tsunami height (over 6 m) was observed only on the sections of the coastline located in the projection of the source on the coast. At the same time, it should be noted that modeling of 1 h of wave propagation in this area (9,600,000 computational nodes) requires less than 1 min of processing time using the proposed FPGA-based calculator.

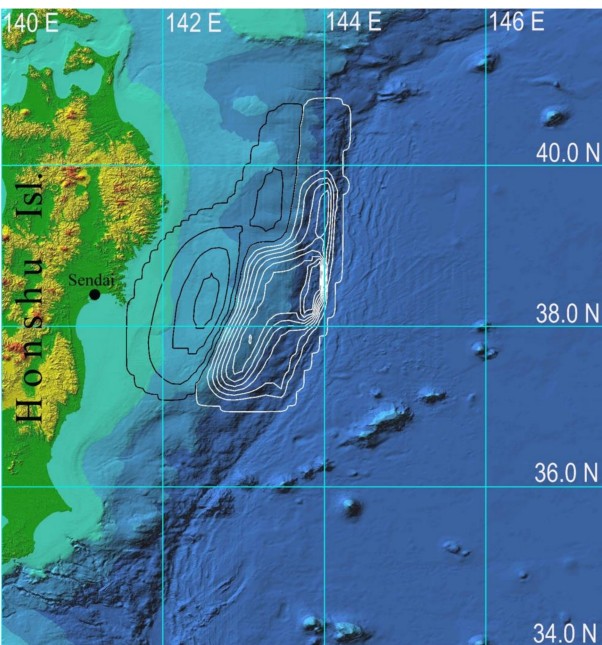

**Figure 4.** Distribution of the tsunami height maxima in the entire computational domain, obtained via numerical modeling using the proposed approach.

## 4. Discussion

This paper shows the advantages of HLS technology, which allows us to significantly simplify and accelerate the development of data processing algorithms. Thus, a relatively insignificant modification of the code of the convolutional algorithm allowed us to construct an effective computational pipeline for the calculation of wave propagation via MacCormack's finite-difference scheme. It should be noted that the application of HLS technology does not require a developer to be professionally acquainted with hardware description languages, such as Verilog; it can also enables hardware paralleling on FPGAs for engineers who are familiar with technologies of parallel programming and know about the structure and features of FPGAs.

As an application of the obtained realization of the MacCormack scheme for numerical solution of the system of shallow water equations, computational experiments were carried out to calculate the maximum heights of tsunami waves generated by a model tsunami source above the linear and parabolic bottom slopes. First of all, it should be noted that the comparison of the results of calculations of the test problems using MOST (the most used tsunami simulation software package worldwide) and the MacCormack scheme implemented on the FPGA with the exact solutions for the model bottom topographies showed their good correspondence. A computational experiment to simulate a catastrophic tsunami on 11 March 2011 off the northeastern coast of Japan (caused by the Great Tohoku Earthquake) gave results similar to those of other authors (e.g., [12]). At the same time, the processing time was an order of magnitude less than even calculations using GPU. This makes it possible to obtain the distribution of the expected tsunami wave heights along the entire coast in 1–2 min, and the use of the nested grids method provides a detailed forecast

for the tsunami-hazardous segments of the coast in which we are interested. The authors expect that the application of the described approaches will make it possible in the future to create a system of local forecasting of the tsunami wave hazard, working in the data entry mode.

It should be noted that GPUs are currently a very popular tool for parallel calculations. In the considered case of the numerical solution of the system of shallow water equations, the performance of the proposed solution based on an FPGA and using a Tesla K-40 GPU was compared. It turned out that the execution time per iteration of the computational algorithm in double precision was 20 ms for the Tesla K-40 GPU and 5.23 ms for the VC709 FPGA. This can be explained by the fact that a special computational pipeline was designed for the use of VC709 FPGA, taking into account the specifics of the proposed implementation of the MacCormack algorithm. This explains the higher performance compared to the standard GPU architecture of the Tesla K-40. Currently, more powerful GPUs—namely, Volta (V100) and Ampere (A100) series—whose peak performance in double precision is 4.94 times better (V100) and 11.72 times better (A100) than the K-40, have been developed. However, the peak performance is usually far from being achieved when solving systems of equations of mathematical physics. In addition, these GPUs are very expensive and server-oriented. The released versions of these GPU series for use in PCs have a reduced memory size and a smaller number of computing cores, while the proposed FPGA-based calculator is designed to work in PCs.

**Author Contributions:** Conceptualization, M.L., M.S. and A.M.; methodology, K.L. and M.S.; software, K.O.; visualization of numerical results—A.M.; writing—original draft preparation, K.O. and A.M.; writing—review and editing, M.L., K.O. and A.M. All authors have read and agreed to the published version of the manuscript.

**Funding:** This research was supported by the state contract with IAE SB RAS (121041800012-8) and with ICMMG SB RAS (0315-2019-0005).

**Institutional Review Board Statement:** Not applicable.

**Informed Consent Statement:** Not applicable.

**Data Availability Statement:** Not applicable.

**Conflicts of Interest:** The authors declare no conflict of interest.

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
