# Peer review of "Algorithmic Design of an FPGA-Based Calculator for Fast Evaluation of Tsunami Wave Danger"

_algorithms, doi:10.3390/a14120343_

Round 1
Reviewer 1 Report
The author present an interesting study of analyzing tsunami waves using FPGA hardware.
The authors discuss the size of the "computational grid" in the abstract and then later in the paper. They should explain in the paper exactly what a computational grid is. Was the size of the grid defined by the problem or was it a choice made by designers? In the abstract the grid is 3120 x 2400. On page 2, the authors refer to a grid of 2580 x 2879 nodes. Is this the same notion of a grid or something different? It is not clear what a node refers to here. Why are the sizes different. These issues need to be more clearly addressed in the paper.
The speed up over a single threaded implementation is good. The authors should provide speedup numbers over a multi-threaded implementation.
The code listings in the paper should be formatted to be more compact.
The paper needs a better discussion of data types used. On the top of page 7 the authors mention replacing floating point types with half precision or fixed point types. They should clearly define what data types are used in the implementation for which they report results. The precision for each design in Table 1 should be clearly defined. The paper should include an error analysis that compares the results of floating point and the reduced precision designs in terms of absolute and relative error.
In table 2, the format for numerical values is not reported.
The results are good. The authors should provide details of the data set analyzed. How large is it. Is it stored in floating point numerical values, and if so, where is the translation from floating point to another format done? How is data translated to an from the FPGA?
Author Response
We are grateful for the Reviewers for their comments and criticism as well as for the time they spent analyzing our paper.
Point 1: Was the size of the grid defined by the problem or was it a choice made by designers? In the abstract the grid is 3120 x 2400. On page 2, the authors refer to a grid of 2580 x 2879 nodes.
Response 1: Grid sizes were corrected to 3000x3200 in both cases, as well as the simulation time. We also add the number of time steps. We add to the Abstract lines 23-25:
“The step length of the computational grid was chosen to display the simulation results in sufficient detail along the coastline. At the same time, the size of the data arrays should provide their free placement of the latter in the memory of each FPGA chip.”
Point 2: The authors should provide speedup numbers over a multi-threaded implementation.
Response 2: We add the required information at the lines 65-69: “This comparison was made against unoptimized, single-threaded reference MOST algorithm running on 3GHz CPU. We have also compared FPGA-based solution against reference MacCormack implementation, parallelized using OpenMP and SIMD instructions, on modern CPU (Intel Core i9-9900K, 8 cores running 4GHz). FPGA-based solution was 11.8 times faster.”
Point 3: The code listings in the paper should be formatted to be more compact
Response 3:
The program listings are formatted as compactly as possible, provided that the perception from the programming point of view is preserved. Further reduction is possible only by switching to algorithmic pseudolanguage, but it contradicts the targeted focus of the journal special issue “Mapping algorithms in FPGA with high-level synthesis”. If acceptable, we would like to show the peculiarities of program writing. Since the size of the article does not exceed the recommended size, we would like to leave the program listings unchanged.
Point 4: The paper needs a better discussion of data types used. On the top of page 7 the authors mention replacing floating point types with half precision or fixed point types. They should clearly define what data types are used in the implementation for which they report results. The precision for each design in Table 1 should be clearly defined. The paper should include an error analysis that compares the results of floating point and the reduced precision designs in terms of absolute and relative error.
Response 4: We add (lines 298) a comment concerning half precision: “However, we do not use this option for tsunami related application.”
We also accompany Table 1 with explanation for precision (lines 308-311): “All the algorithms at any platform operates on data in single-precision float-point (IEEE 754), no conversions to fixed-point or half-precision were made. Such conversion may result in better FPGA resource utilization, but it requires deep data analysis for every case and is beyond our consideration here.”
Point 5: In table 2, the format for numerical values is not reported.
Response 5: We believe that the explanations of Response 4 addresses also the Table 2.
Point 6: The results are good. The authors should provide details of the data set analyzed. How large is it. Is it stored in floating point numerical values, and if so, where is the translation from floating point to another format done? How is data translated to an from the FPGA?
Response 6: We add explanations right after the Table 2 (lines 363-371):
“Once again, we would like to stress that all the algorithms operate on data in single-precision float-point (IEEE 754), no conversions to fixed-point or half-precision were made. The entire data array (D, η, u, v) is stored in dynamic RAM (DDR), placed near FPGA chip on board. Initial data transferred to FPGA from PC using PCIe interface, or internal interconnect in case of FPGA+CPU solution, such as Xilinx Zynq. Maximum acceptable data set size depends on the size of dynamic RAM. For VC709 accelerator, maximum dataset is 16384x16384 points of bathymetry. Bathymetry sizes for our experiments is 3000x3200 for practical reasons (size of resulting data, computation time on FPGA/CPU, and other).”
In addition, for better explanations of the Figures 2(a) and 2(b) we introduce the lines 401-406:
“The propagation of the wave generated by the described round-shaped source was simulated using MOST software, and by the McCormack difference scheme implemented on FPGA. According to the results of calculations for each of the schemes, isolines of distributions of wave heights maxima over the sloping bottom (Figure 2(a)) and at the parabolic bottom topography (Figure 2(b)) were visualized.”

Reviewer 2 Report
This paper discusses the design and high-level synthesis of an FPGA accelerator tailored to fast evaluation of tsunami wave danger. The paper is well written and is easy to follow but it lacks some important points which should be corrected:
- “An acceleration of about 300 times has been achieved compared to single-threaded computing.” Please provide details explaining this result. On which platform has this acceleration been achieved? Which interface was used for supplying input data? What kind of “single-thread computer” was used for comparison?
- The authors mention that a “hardware-software solution” is proposed, but the results section does not explain what tasks are mapped to software and what tasks are mapped to hardware (and what kind of communication channel is organized).
- “For Verilog code, you need to modify the code to increase the frequency.” – This is only partially true, as well as this is partially true for HLS. To increase the frequency you might set tighter timing constraints in XDC file and re‐run synthesis/implementation until WNS<0.
- “All solutions were analyzed for maximum frequency using Vivado tools.” What particular technique have the authors use to get this parameter? As you know, the maximum frequency value is not explicitly given in Vitis/Vivado timing reports.
- “functions in differential system (1)” – where can we see the system (1)?
- Provide explanations on data from Table 2. Which language did you use for synthesis? According to these data, it seems that more LUTs are used than available on the indicated board.
- “VC709 FPGA chip” – there is no FPGA cheap with this name on the market. Are you referring to a VC709 board?
- In all the experiments, it is not clear how the data have been supplied.
- How would you explain acceleration of your platform comparing to GPU clusters?
- Please remove the following text:
“References must be numbered in order of appearance in the text (including citations in tables and legends) and listed individually at the end of the manuscript. We recommend preparing the references with a bibliography software package, such as EndNote, ReferenceManager or Zotero to avoid typing mistakes and duplicated references. Include the digital object identifier (DOI) for all references where available.”
- Typos: “selected cased” -> cases; “for the a distance” -> the
Author Response
We are grateful for the Reviewers for their comments and criticism as well as for the time they spent analyzing our paper.
Point 1: “An acceleration of about 300 times has been achieved compared to single-threaded computing.” Please provide details explaining this result. On which platform has this acceleration been achieved? Which interface was used for supplying input data? What kind of “single-thread computer” was used for comparison?
Response 1: We add the required information at the lines 65-69: “This comparison was made against unoptimized, single-threaded reference MOST algorithm running on 3GHz CPU. We have also compared FPGA-based solution against reference MacCormack implementation, parallelized using OpenMP and SIMD instructions, on modern CPU (Intel Core i9-9900K, 8 cores running 4GHz). FPGA-based solution was 11.8 times faster.”
Point 2: The authors mention that a “hardware-software solution” is proposed, but the results section does not explain what tasks are mapped to software and what tasks are mapped to hardware (and what kind of communication channel is organized).
Response 2: We add some explanations, lines 73-77: “To clarify the use of hardware-software terminology, we note that all computations were mapped to hardware, software is used to organize access to data and save results. Data is stored on dynamic RAM (DDR3/4) attached to FPGA. Communication is done through internal interconnect in case of Zynq SoC, or PCIe in case of VC709, using DMA engine on FPGA and driver software on host.”
Point 3: “For Verilog code, you need to modify the code to increase the frequency.” – This is only partially true, as well as this is partially true for HLS. To increase the frequency, you might set tighter timing constraints in XDC file and re‐run synthesis/implementation until WNS<0.
Response 3: We agree, so, the “justification” paragraph has been added, lines 290-295: “In fact, there are alternative options to increase the frequency. However, synthesis and implementation are very strict in what they can done with design because design structure/behavior is strictly specified. For example, they cannot increase pipeline depth automatically to achieve better timing. HLS tools on the other side are free in choosing way to implement desired algorithm.”
Point 4: “All solutions were analyzed for maximum frequency using Vivado tools.” What particular technique have the authors use to get this parameter? As you know, the maximum frequency value is not explicitly given in Vitis/Vivado timing reports.
Response 4: We agree, explanations are added, lines 302-307:
“Strictly speaking, the maximum frequency is not explicitly given in Vitis/Vivado timing reports. Vivado HLS can report Fmax, calculated from post-synthesis timing reports (using “target period - worst slack” formula), same was done for Verilog design. Target frequency (specified in constraints) is same as target frequency for HLS design (400MHz and 148.5MHz) and 148.5MHz for Verilog. That’s not a real frequency on what implemented design can run.”
Point 5: “functions in differential system (1)” – where can we see the system (1)?
Response 5: The system (1) had been inserted at lines 327-335.
Point 6: Provide explanations on data from Table 2. Which language did you use for synthesis? According to these data, it seems that more LUTs are used than available on the indicated board.
Response 6: The following details were added, lines 357-360:
“Below we present an implementation of MacCormack algorithm in C using HLS. Design is targeted for ZCU106 board (FPGA xczu7ev) and 400MHz. Xczu7ev has 230400 LUTs, 460800 flip-flops, 312 + 512 BRAM blocks (36K) and 1728 DSP blocks.”
Point 7: “VC709 FPGA chip” – there is no FPGA cheap with this name on the market. Are you referring to a VC709 board?
Response 7: Sorry, we made a correction at line 23
Point 8: In all the experiments, it is not clear how the data have been supplied.
Response 8: We add explanations right after the Table 2 how is data translated to and from the FPGA (lines 363-371):
“Once again, we would like to stress that all the algorithms operate on data in single-precision float-point (IEEE 754), no conversions to fixed-point or half-precision were made. The entire data array (D, η, u, v) is stored in dynamic RAM (DDR), placed near FPGA chip on board. Initial data transferred to FPGA from PC using PCIe interface, or internal interconnect in case of FPGA+CPU solution, such as Xilinx Zynq. Maximum acceptable data set size depends on the size of dynamic RAM. For VC709 accelerator, maximum dataset is 16384x16384 points of bathymetry. Bathymetry sizes for our experiments is 3000x3200 for practical reasons (size of resulting data, computation time on FPGA/CPU, and other).”
Point 9: How would you explain acceleration of your platform comparing to GPU clusters?
Response 9: We add our vision of the required explanations as the last paragraph in the Discussion, lines 461-476:
“Note that GPUs are currently a very popular tool for parallel calculations. In the considered case of numerical solution of the system of shallow water equations, the performance of the proposed solution based on FPGA and using Tesla K-40 GPU was compared. It turned out that the execution time per iteration of the computational algorithm in double precision was 20 ms for the Tesla K-40 GPU and 5.23 ms for the VC709 FPGA. This can be explained by the fact that a special computational pipeline has been designed for the use of VC709 FPGA. It takes into account the specifics of the proposed implementation of McCormack algorithm. This explains the higher performance compared to the standard GPU architecture of Tesla K-40. Currently the more powerful GPUs, namely Volta (V100) and Ampere (A100) series, whose peak performance in double precision is 4.94 times better (V100) and 11.72 times better (A100) than the K-40, have been developed. However, the peak performance is usually far from being achieved when solving systems of equations of mathematical physics. In addition, these GPUs are very expensive and server-oriented. The released versions of these GPU series for use in the PC have a reduced memory size and a smaller number of computing cores, while the proposed FPGA-based calculator is designed to work in the PC.”
Point 10: Please remove the following text: “References must be numbered….”
Response 10: Sorry for that, template instructions were removed.
Point 11: Typos: “selected cased” -> cases; “for the a distance” -> the
Response 11: Word “selected” in Abstract was changed by “two particular”, mistake “the a” was corrected.
In addition, for better explanations of the Figures 2(a) and 2(b) we introduce the lines 401-406:
“The propagation of the wave generated by the described round-shaped source was simulated using MOST software, and by the McCormack difference scheme implemented on FPGA. According to the results of calculations for each of the schemes, isolines of distributions of wave heights maxima over the sloping bottom (Figure 2(a)) and at the parabolic bottom topography (Figure 2(b)) were visualized.”

Round 2
Reviewer 1 Report
The authors have addressed my comments in this revision.
Reviewer 2 Report
I am satisfied with the revised version of the manuscript. I would only suggest to define what particular internal interconnect was used in the following case:
"Communication is done through internal interconnect in case of Zynq SoC"